# Methyl Orange Biodegradation by Immobilized Consortium Microspheres: Experimental Design Approach, Toxicity Study and Bioaugmentation Potential

**DOI:** 10.3390/biology11010076

**Published:** 2022-01-05

**Authors:** Amany Ibrahim, Esmail M. El-Fakharany, Marwa M. Abu-Serie, Marwa F. ElKady, Marwa Eltarahony

**Affiliations:** 1Botany Department, Faculty of Women for Arts, Science and Education, Ain Shams University, Cairo 11566, Egypt; 2Department of Biology, College of Science, Taif University, P.O. Box 11099, Taif 21944, Saudi Arabia; 3Protein Research Department, Genetic Engineering and Biotechnology Research Institute (GEBRI), City of Scientific Research and Technological Applications (SRTA-City), New Borg El-Arab City, Alexandria 21934, Egypt; 4Medical Biotechnology Department, Genetic Engineering and Biotechnology Research Institute (GEBRI), City of Scientific Research and Technological Applications (SRTA-City), New Borg El-Arab City, Alexandria 21934, Egypt; marwaelhedaia@gmail.com; 5Chemical and Petrochemical Engineering Department, Egypt-Japan University for Science and Technology, New Borg El-Arab City, Alexandria 21934, Egypt; marwa.f.elkady@gmail.com; 6Fabrication Technology Researches Department, Advanced Technology and New Materials Research Institute, City of Scientific Research and Technological Applications (SRTA-City), New Borg El-Arab City, Alexandria 21934, Egypt; 7Environmental Biotechnology Department, Genetic Engineering and Biotechnology Research Institute (GEBRI), City of Scientific Research and Technological Applications (SRTA-City), New Borg El-Arab City, Alexandria 21934, Egypt

**Keywords:** azo dye degradation, alginate immobilization, central composite design, reductases, consortium microspheres

## Abstract

**Simple Summary:**

An efficient immobilized bacterial consortium of *Raoultella planticola*, *Ochrobactrum thiophenivorans*, *Bacillus flexus*, and *Staphylococcus xylosus* was used to degrade MO under high salinity conditions. The biodegradation efficiency was confirmed by UV−visible spectrophotometric analysis, FTIR, and the assaying of degradative enzymes. The cytotoxicity of such metabolic byproducts was evaluated on two human normal cell lines. In addition, the immobilized consortium showed the synergistic interaction between immobilized consortium and indigenous microorganism for degrading MO in artificially contaminated agricultural and industrial effluents.

**Abstract:**

Methyl orange (MO) is categorized among the recalcitrant and refractory xenobiotics, representing a significant burden in the ecosystem. To clean-up the surrounding environment, advances in microbial degradation have been made. The main objective of this study was to investigate the extent to which an autochthonous consortium immobilized in alginate beads can promote an efficient biodegradation of MO. By employing response surface methodology (RSM), a parametric model explained the interaction of immobilized consortium (*Raoultella planticola*, *Ochrobactrum thiophenivorans*, *Bacillus flexus* and *Staphylococcus xylosus*) to assimilate 200 mg/L of MO in the presence of 40 g/L of NaCl within 120 h. Physicochemical analysis, including UV-Vis spectroscopy and FTIR, and monitoring of the degrading enzymes (azoreductase, DCIP reductase, NADH reductase, laccase, LiP, MnP, nitrate reductase and tyrosinase) were used to evaluate MO degradation. In addition, the toxicity of MO-degradation products was investigated by means of phytotoxicity and cytotoxicity. *Chlorella vulgaris* retained its photosynthetic performance (>78%), as shown by the contents of chlorophyll-a, chlorophyll-b and carotenoids. The viability of normal lung and kidney cell lines was recorded to be 90.63% and 99.23%, respectively, upon exposure to MO-metabolic outcomes. These results reflect the non-toxicity of treated samples, implying their utilization in ferti-irrigation applications and industrial cooling systems. Moreover, the immobilized consortium was employed in the bioremediation of MO from artificially contaminated agricultural and industrial effluents, in augmented and non-augmented systems. Bacterial consortium remediated MO by 155 and 128.5 mg/L in augmented systems of agricultural and industrial effluents, respectively, within 144 h, revealing its mutual synergistic interaction with both indigenous microbiotas despite differences in their chemical, physical and microbial contents. These promising results encourage the application of immobilized consortium in bioaugmentation studies using different resources.

## 1. Introduction

The genetic diversity and wealthy nature of microbial metabolism are considered the paramount agents in its governing several vital medical, pharmaceutical, agricultural, environmental and industrial applications. At the environmental level, microbial metabolism was recruited in resolving numerous environmental issues. Through bioremediation, either bacterial [1] or mycoremediation [2], several man-made complicated xenobiotic compounds are degraded to simpler inorganic constituents or even mineralized into water and carbon dioxide. Such a naturally occurring phenomenon offers many advantages over physical and chemical methods, including eco-friendliness, cost-effectiveness and less sludge generation or incomplete degraded byproducts, which represent additional environmental burdens [3,4]. However, the effective implementation of biodegradation processes depends mainly on the behavior of microbes toward xenobiotic compounds via the secretion of adequate enzymatic batteries or non-enzymatic redox-active molecules. Actually, the employing of advanced trials in molecular, bioinformatics, biotechnological and even basic microbiological applications, either segregated or in alliance, aid in accelerating the biodegradation process efficiently. Interestingly, genetic transformation, bio-stimulation, bio-augmentation, mixed cultures and immobilization are among the attractive techniques pertaining to the proficient microbial degradation of recalcitrant substances [5,6]. Bioaugmentation is the addition of exogenous microorganisms, characterized by their biodegradation capacities, to indigenous or in situ microbial populations for improving the treatment process of pollutants [7]. The success in this task is mostly based on the proliferative and competitional proficiency of the introduced microbes [8]. However, the comprehensive understanding of addition/maintenance techniques, and knowledge of molecular biology and the capabilities of commercial products, are prerequisites [9]. Additionally, adjusting and monitoring some parameters such as temperature, pH, dissolved oxygen, nutrition availability, toxicity and microbial pressures during the treatment process contributes to its success; otherwise, it fails, as revealed in [10].

Notably, the employment of statistical approaches to the design of experiments (DOEs) such as central composite design (CCD) in the remediation process, either biologically or chemically, has recently become more common [11,12,13]. Through these chemometric means, the optimum conditions of multiple independent variables can be defined. Importantly, the interaction effect of each examined variable on the response would be determined without exerting more effort, with more time saved and the process being more cost-effective [14,15]. Interestingly, the utilization of CCD in chemical processes such as wastewater flocculation and coagulation, protein extraction from wastewater and dye degradation was revealed [16]

Remarkably, the comparison of the biodegradative performance of monocultures and mixed cultures always favors the mixed cultures, as evidenced by Ahmadi et al. [17] and Phan et al. [18]. Microbial consortia demonstrate stability in the cleavage of synthetic contaminants; the metabolic burden of synthetic pollutant degradation is divided across multiple microbes, and each microbe in the consortium participates with its own mechanism in the degradation process [19]. Such combinational action of all the consortium’s microbial partners speeds up the biodegradation rate [20]. Additionally, the microbes in their mixed system seem to be resistant to abiotic stressors. A number of recent studies listed the enhanced impact of different microbial consortiums in the removal of a wide array of pollutants, including heavy metals [21], organochlorine pesticide [22], organophosphate pesticides [23], azo dye [24] and organic hydrocarbons [25,26]. On the other hand, the use of microbial biomass in the form of an immobilized state is advantageous as compared to the freely suspended state [27]. This could be explained by the protection of the immobilized cells from the toxic impact of pollutants or their byproducts, and their stability for reuse. Thus, the immobilization technique is considered as a powerful tool that can be utilized towards various applications in an economic manner [28,29]. The entrapment approach is used in the overwhelming majority of immobilization techniques, especially in alginate beads. It is a natural polymer, and is cheap, nontoxic, biocompatible to microbial cells, readily available, highly viscous and exhibits good stability. Furthermore, its gel nature furnishes suitable porosity, allowing the continuous diffusion of substrates and byproducts [30]. In broader terms, the utilization of alginate gel immobilized with microbial biomass was recorded in different applications, such as alcoholic wine fermentation [31], dye decolorization [32], crack remediation [33] waste oil degradation [34] and heavy metal removal [24,35].

The combination of such approaches in the degradation of recalcitrant substrates generated from anthropogenic activities, such as azo dye, seems to be promising [36,37]. Actually, azo dye represents about almost 80% of synthetic dyes that enter water bodies from textile discharge [38], causing environmental injuries such as reductions in dissolved oxygen concentrations, the blocking of sunlight penetration, and the blocking of photosynthesis [39,40,41]. These negative effects are compounded by its adverse impact on human health, including respiratory problems, diarrhea, vomiting and nausea [42]. The continuous presence of azo dyes in ecosystems further leads to poisoning, carcinogenicity, teratogenicity and mutagenicity [43]. It is worth mentioning that textile discharge is characterized by its salty nature, as stated by Mirbolooki et al. [44]. In fact, higher concentrations of NaCl are added during the staining process to neutralize the surface charge of negatively-charged fibers; hence, the fixation and adsorption of anionic dyes can be improved. By these means, the leakage of considerable quantities of salt along with dyestuff takes place, which varies from 1 to 10 g/L, causing further encumbrance.

On this basis, the current study focused on the biodegradation of Methyl orange (MO), as one of the most extensively used hazardous anionic azo dyes [45]. Methyl orange is an azo dye that has been sulphonated [46]. In laboratories, it is also used as a pH indicator. It is brightly colored and highly soluble in water, making it difficult to remediate using traditional treatment methods (physicochemical methods) [47]. Its toxic nature and resistance to degradation necessitate the discovery of more azo dye-degrading microorganisms. A statistical experiment design was employed to study the interactive effect of some variables on the degradation performance of the immobilized microbial consortium. Moreover, the activity of some oxidoreductive enzymes, which are responsible for biodegradation, were quantitatively detected during the process. In addition, UV-Visible spectra and FTIR analysis were employed to confirm the breakdown and detoxification of MO. Furthermore, the acute toxicity assay, which encompassed a phytotoxicity assessment, was conducted, with the results being evaluated against those obtained for algae. Additionally, the cytotoxicity of the degraded products, as compared to the MO dye, was determined on human and animal cell lines. Additionally, a bioaugmentation study of the consortium immobilized in alginate beads was conducted on real effluents that were artificially contaminated with MO in comparison to free consortium.

## 2. Materials and Methods

### 2.1. Screening, Selection and Growth Condition Analysis of MO-Degrading Microorganisms

In this article, strains from the in-house culture collection isolated from various environmental sources (sediments and fresh water) and industrial effluents were examined for degradation of MO. The coordinates of the screening sites were as follows: Naba Alhamra, Wadi El Natrun (morphotectonic depression at the northeast corner of the Western Desert between the longitudes of 30°00′ and 30°30′ E and the latitudes of 30°15′ and 30°40′ N, Al-Beheira governorate, Egypt), El-Mahmoudeya Canal (45-mile-long (72 km) sub-canal derived from the Nile River and influx into the Mediterranean Sea, latitude 31°11′2″ N and longitude 30°31′27″ E, Alexandria governorate, Egypt) and Jeddah’s Second Industrial City neighborhood (latitude 21°48′ N and longitude 39°19′ E, Saudi Arabia). The isolated strains were directly screened for MO degradation using a plate test [48]. Using mineral salt medium (MSM) (g/L) (1.27 K_2_HPO_4_; 0.42 MgSO_4_·7H_2_O; 0.42 NaNO_3_; 0.29 KCl; 2 NaCl; 0.85 KH_2_PO_4_, pH 7.0) supplemented with 50 mg/L of MO, constant microbial lawns were spotted separately on the surface of the solidified agar plates and incubated for 72 h at 30 °C. A discoloration zone was observed around some colonies, which demonstrated their preliminary ability for degradation. The colonies with the most discolored areas, measured from the colony’s edge, were selected to construct the consortium.

### 2.2. Antagonism Assay for Detecting the Compatibility of the Consortium Strains

The bacterial compatibility was evaluated for consortia as described by [49]. Briefly, about 50 μL of each bacterial suspension (about 10^6^ (CFU) mL^−1^) was spotted onto nutrient agar plates (g/L) (2 yeast extract; 5 peptone; 5 sodium chloride; 15 agar, pH 7.0) with 1 cm spaces between the spots. The inoculated plates were then incubated at 30 °C in a static condition for 72 h. The overlapping of the spots implied that there was compatibility without symptoms of inhibition being shown [50].

### 2.3. Inoculum Preparation and Consortium Immobilization in Alginate Microspheres

A pure culture of each bacterial strain was inoculated in NB and incubated at 30 °C in an orbital shaker (150 rpm) (MCL-D21 Rotary shaker) until the OD_600_ reached 0.6 to 0.8. The cultures were then collected aseptically and centrifuged at 10,000 rpm for 10 min. At equal cell densities, the harvested pellets were resuspended in distilled water and homogeneously mixed. The consortium was immobilized by entrapment in a 3% sodium alginate solution following the method described by Eltarahony et al. [51]. A quantity of 50 mL of bacterial biomass was mixed thoroughly with a slurry of 3% sodium alginate (50 mL) in a 250 mL Erlenmeyer flask. The homogeneous mixture was dropped into 100 mL of 2% chilled sterile CaCl_2_ (*w*/*v*) using a syringe, thereby forming gel beads (~4 mm diameter). The immobilized consortium beads were kept in CaCl_2_ (2%) for 1 h. at 30 °C to complete the hardening process. Thereafter, the collected beads were washed several times using sterile distilled water to remove any remains before freeze-drying for 8 h at 36 °C [52,53].

### 2.4. Characterization of Immobilized Beads by Scanning Electron Microscopy (SEM)

SEM (JSM 6360LA, JEOL, Akishima-shi, Japan) was used to examine the morphology of the beads. Some beads (before and after bacterial immobilization) were dehydrated at 30 °C using a vacuum stove until constant weight was achieved, and hydrogels were freeze-dried for 8 h at −20 °C and 0.022 mmHg using a Rificor S.H lyophilizer. Dehydrated and lyophilized beads were cryo-fractured in liquid nitrogen before being mounted on bronze stubs and metallized with an argon plasma metallizer (PELCO 91000 sputter coater) [53].

### 2.5. Biodegradation Test of Immobilized Beads

MO degradation for immobilized consortium was screened initially in a 100 mL MSM broth (mentioned previously) supplemented with 100 mg/L MO and 7 immobilized beads containing 10^6^ CFU/mL. All tests were performed in triplicates and samples were incubated for 7 days at 30 °C in a rotary shaker set to 150 rpm (MaxQ 6000, Thermo Fisher, Waltham, MA, USA). At time intervals of 6 h throughout the incubation period, the MO residues were measured using a spectrophotometer at wavelength of 465 nm [54,55].

### 2.6. MO Degradation by Consortium Immobilized in Alginate Beads Using Central Composite Design (CCD)

In this step, the biodegradation performance was examined simultaneously and statistically in relation to certain variables [56]. The statistical modeling of the degradation solution was constructed based on 5 experimental levels, −*α*, −1, 0, +1, and +*α*, for four independent variables (degradation time, inoculated consortium, MO and NaCl concentrations) in a 31-trail matrix. The experimental levels of the examined independent variables are illustrated in Table 1.

The statistical calculation that describes the relationship between the coded and actual values is represented in Equation (1):*Xi* = *Ui* − *Ui*_0_/Δ*Ui*(1)
where *Xi* is the coded value of the *i*th variable, *Ui* is the actual value of the *i*th variable, *Ui*_0_ is the actual value of the *i*th variable at the center point and Δ*Ui* is the step change of the variable. The relationship between the examined independent variables and the response (MO detoxification) was calculated using the equation of second-degree polynomial (Equation (2)):*Y = β*_0_ + *β*_1_*X*_1_ + *β*_2_*X*_2_ + *β*_3_*X*_3_ + *β*_11_*X*_11_+ *β*_22_*X*_22_ + *β*_33_*X*_33_ + *β*_12_*X*_1_*X*_2_ + *β*_13_*X*_1_X_3_ + *β*_23_*X*_2_*X*_3_
(2)
where: *Y* is the predicted response; *X*_1_, *X*_2_ and *X*_3_ are input variables that influence the response variable *Y; β*_0_ is the intercept; *β*_1_, *β*_2_ and *β*_3_ are linear coefficients; *β*_11_, *β*_22_ and *β*_33_ are squared or quadratic coefficients, and *β*_12_, *β*_13_ and *β*_23_ are interaction coefficients.

### 2.7. Statistical Analysis

The experimental designs, the regression analysis for determining the analysis of variance (ANOVA), three-dimensional surface plots (3D) and two-dimensional contour plots (2D) were established using the Minitab 14.0 statistical software (Minitab Inc., State College, PA, USA).

### 2.8. Decolorization Assay of MO

For each CCD-trial, MO decolorization was assessed by determining the change in absorbance of clear supernatants at the absorption maxima (λ max) of 465 nm with a UV–Vis spectrophotometer (Labomed model, Inc., Los Angeles, CA, USA). Abiotic control or un-inoculated medium was prepared in the CCD matrix for each trial. The experiments were performed in triplicate and the average was considered. The color removal efficiency was calculated as the percentage ratio based on the following equation [49,57]:(3)$$\mathrm{Dye}\;\mathrm{Degradation}\;\left(\%\right)=\frac{\mathrm{Initial}\;\mathrm{OD}-\mathrm{Final}\;\mathrm{OD}}{\mathrm{Initial}\;\mathrm{OD}}\;\times \;100.$$

#### 2.8.1. Extraction and Analysis of Degradation By-Products

Both MO and its degraded metabolites were subjected to further analysis to confirm the accomplishment of the biodegradation process. The immobilized consortium was inoculated in MSM broth containing 200 mg/L of MO and incubated at 30 °C in a 150 rpm orbital shaker for 5 days. Simultaneously, a control experiment was conducted in exactly the same manner as outlined previously but without consortium inoculum. At the end of the incubation time, a 50 mL quantity of the MO (control dye without degradation) and the degraded sample was centrifuged at 10,000 rpm for 10 min. The obtained supernatants were extracted using ethyl acetate in equal proportion followed by evaporation by rotary evaporator (IKA Rotary evaporator RV 8 V-C). The generated extract was suspended in water (5 mL) for the further analysis.

#### 2.8.2. UV−Visible Spectrophotometric Analysis

A UV–Visible spectrophotometer (Labomed model, Inc., USA) was used to assess biodegradation at the zero time point and after the completion of the process. A clear supernatant was scanned from 200 to 800 nm within the absorption spectrum.

#### 2.8.3. FTIR

The FTIR spectra of MO and its degraded metabolites were determined using the Shimadzu FTIR-8400S (Kyoto, Japan). Dehydrated and lyophilized samples were triturated into fine powder, combined with KBr (1.0 percent *w*/*w*) and crushed into transparent discs in a hydraulic press. Spectra were captured at 4 cm^−1^ resolution over a range of 4000–400 cm ^−1^ with an accumulation of 64 scans and using dried air as a background [58].

### 2.9. Enzymes Activity Assays

The degradation performance of the bacterial consortium was examined at flask level in MSM broth supplemented with 200 mg/L of MO (i.e., the highest concentration was degraded efficiently by the consortium). About 10^6^ CFU mL^−1^ was utilized as an inoculum and incubated as mentioned previously. The activities of the enzymes, including manganese peroxidase (MnP), nitrate reductase (NR), tyrosinase, lignin peroxidase (LiP), NADH-DCIP reductase, azoreductase and laccase, were quantified during the biodegradation process at a constant interval time (12 h). The crude enzymes were prepared by cell lysis using the physical disruption method (Ultrasonic Disruptor UD-200, Tommy, Tokyo, Japan) for 2 min intervals based on a 40-amplitude output at 4 °C. The obtained slurry was subsequently centrifuged for 5 min at 4 °C and 15,000 rpm, and the clear solution was used as a crude enzyme. The enzyme assays were determined as reported by Glenn and Gold [59], Redinbaugh and Campbell [60], Shanmugam et al. [61], Sambasiva Rao et al. [62], Zhao et al. [63], El-Fakharany et al. [64], Rekik et al. [65], and Sari and Simarani [66,67] Negative controls (enzymes denatured by boiling) were run in parallel.

### 2.10. Protein Content Determination

The protein content in all tested samples was determined by reading the OD values of samples at 280 nm or using the method outlined in [68] with BSA as a standard protein.

### 2.11. Acute Toxicity Assessments

#### 2.11.1. Phytotoxicity Bioassays with *Chlorella vulgaris*

The toxic effects of MO (200 mg/L) before degradation and its byproduct (after degradation) were compared with control (without any treatment). In three Erlenmeyer flasks (500 mL), the *C. vulgaris* cells were inoculated into 100 mL of Bold’s basal media (BBM) medium alone and in combination with 200 mg/L of either MO or the degradation byproducts. All flasks were incubated for 4 days at 25 °C under illumination [69]. Following the completion of the incubation period, a fixed volume of algal cultures was centrifuged at 3000 rpm for 20 min. The obtained biomass was subjected to pigment fractionation in accordance with Shebany et al. [70]. Pigments were extracted in hot methanol (70 °C) for 10 min. Centrifugation (Sorvall RC-3C Plus refrigerated floor model) was used to remove cell debris, and the clear supernatant, which contained the pigments, was aspirated and diluted to a specific volume. At 452.5, 644 and 663 nm, the extinction coefficient was measured using a spectrophotometer against a methanol blank. Taking dilution into account, the pigment fractions’ contents (µg/mL algal suspension) were calculated using the following equations [70]:(4)Chlorophyll a =10.3 E663−0.918 E644 
(5)Chlorophyll b =19.7 E644−3.87 E663 
(6)Carotenoids =4.2 E452.5−[0.0264 Chl. a +0.4260 Chl. b]

#### 2.11.2. Determination of Cytotoxicity against Normal Lung and Kidney Cell Lines

A human normal lung fibroblast cell line (Wi-38) and a normal adult African green monkey kidney cell line (Vero) were used to investigate the toxicity of MO dye before and after bioremediation according to the tetrazolium salt (MTT) assay [71]. The MTT assay was performed to detect the viability of the treated cells as compared to healthy untreated cells. Only viable active metabolic cells have the ability to reduce yellow soluble tetrazolium salt to insoluble purple formazan by means of their mitochondrial dehydrogenase enzyme. Human Wi-38 and mammalian Vero cells were maintained in Dulbecco’s Modified Eagle (DMEM) medium (Lonza, Bend, OR, USA) containing 10% fetal bovine serum. These cell lines were sub-cultured for 2 weeks before an assay was conducted using trypsin EDTA (Lonza, USA). Their viability and counting were detected by trypan blue staining and with the use of a hemocytometer. Wi-38 and Vero cells were seeded in a 96-well culture plate at 1 × 10^4^ cells per well and incubated at 37 °C in a 5% CO_2_ incubator. After 24 h, for the cell attachment, the cells were exposed to MO (200 mg/L) and its degraded byproducts. Following 72 h of incubation in a 5% CO_2_ incubator, 20 µL of MTT solution (3-(4,5-Dimethylthiazol-2-yl)-2,5-Diphenyltetrazolium Bromide) (5 mg/mL) was added to each well before incubation at 37 °C for 4 h in a 5% CO_2_ incubator. MTT (Sigma, St. Louis, MO, USA) solution was removed and the insoluble blue formazan crystals trapped in cells were solubilized with 150 µL of 100% DMSO at 37 °C for 10 min. The absorbance of each well was measured with a microplate reader (BMG LabTech, Ortenberg, Germany) at 570 nm to estimate cell viability.

### 2.12. Bioaugmentation of Free and Immobilized Cells in Real Waste Effluents

The biodegradative potential of immobilized consortium beads in real wastewater containing various pollutant loads was determined in this step. In comparison to free non-immobilized consortium, the immobilized beads were employed as a modular bioaugmentation tool for remediating two effluents that were artificially inoculated with MO (200 mg/L) in two forms, namely sterilized (non-augmented) and non-sterilized (augmented) states. The industrial effluent was collected, in December 2020, from Discharge Line Industrial Zone, Borg Al Arab, Alexandria, Egypt, while the agricultural effluent was collected from Bahig Canal, Borg Elarab, Alexandria, Egypt, in October 2020. The quality criteria of the samples were identified and their quality was measured according to these standards. In parallel, non-immobilized alginate beads were run to detect the absorption percentage. For bench-scale bioremediation trials, fixed numbers of immobilized alginate beads, of equivalent size to the free consortium, and non-immobilized alginate beads were inoculated in 1 L flasks containing 600 mL of industrial and agricultural effluents in both augmented and non-augmented states. The incubation conditions were performed as recorded previously. At each time interval (24 h), 2 mL quantities of samples were drawn, centrifuged and measured spectrophotometrically at 465 nm. The biodegradation percentage was calculated using Equation (3) [72].

## 3. Results

### 3.1. Selection of MO-Degrading Microorganisms and Their Compatibility

The potential of 39 bacterial strains to degrade MO was examined; four of them were selected based on the highest values of the clearance zone that surrounded their colonies. The selected strains were: *Raoultella planticola*, *Ochrobactrum thiophenivorans*, *Bacillus flexus* and *Staphylococcus xylosus*. Their 16S-rDNA genes were identified and deposited in *GenBank* under accession numbers MK551748, MN631047, MT940225 and MT940226, respectively. Such strains were isolated from different ecosystems that contained different pollutant loads. They exhibited characteristic biochemical and physiological capabilities and some of them were utilized previously in the bioremediation of heavy metals [58]. As noticed, the selected strains are affiliated to different bacterial classes encompassing *alphaproteobacteria*, *gammaproteobacterial and firmicutes*, which imply different behaviors in terms of the handling of substrates. However, their combination in the consortium depends mainly on the absence of interspecies antagonism, which was clearly demonstrated in the compatibility experiment. Such corporative interactions facilitate their application, as noted by Fuentes et al. [73]. On highly nutritive media such as nutrient broth, MO was easily and entirely degraded in time range of 20–28 h by four selected strains as monocultures; upon combination into a mixed culture, removal was achieved within 18 h (data not shown). Nonetheless, the reason behind utilizing MSM in the present study is based on its capacity to resemble reality and mimic the effluents that are full of stressors and have a lack of nutrients. Herein, MO was assimilated by the examined consortium and 30% of it was eliminated during 7 days of incubation at the basal MSM, where it consumed MO as a carbon source and attacked its structure to obtain enough energy for survival and proliferation. Individually, in terms of discoloration, *Raoultella planticola*, *Ochrobactrum thiophenivorans*, *Bacillus flexus* and *Staphylococcus xylosus* showed 7, 9, 13 and 16% color removal, respectively, in MSM within the exact conditions, reflecting the enhancement of degradation by the action of combinatorial configuration. The porous structure of alginate microsphere that granted the migration of the dye’s molecules from the solution into the bulk structure of microspheres is shown in Figure 1A. Moreover, successful entrapment of the oval shaped bacterial consortium cells onto the porous alginate microsphere matrix was indicated via SEM (Figure 1B). It is evident from this figure that the porous structure of alginate enables the interior migration of MO dye’s molecules onto the alginate microspheres, which allows them to interact with bacterial strains, thereby facilitating efficient dye degradation.

### 3.2. MO Degradation by Consortium Immobilized in Alginate Beads Using Central Composite Design (CCD)

To study the degradation behavior of immobilized consortium with different inoculated bacterial loads, in response to different ranges of MO and NaCl concentrations, over different incubation time periods, CCD was employed in 31 experimental runs. To our knowledge, this has rarely been investigated to date. The design matrix, which was composed of 16 factorial points (cubic points), 8 axial or star points (points with an axial distance to the center of (*a* = ±2)), and 7 replicates of center points, is represented, along with experimental, predicted responses and the studentized residual results, in Table 2. As observed, the highest percentage of MO degradation, at 46.38%, was recorded for trial 2 (axial point); however, the minimum removal percentage (0.27%) was noted for trial 6 (axial point), reflecting the interaction effect of different parameters at varied levels on response.

### 3.3. Multiple Regression Analysis and ANOVA

The multiple regression analysis (Table 3) and ANOVA (Table 4) methods were employed to analyze the CCD results. By these means, the individual model coefficients, the significance of the regression model, the lack-of fit, the coefficient values of each examined parameter, their probability *p*-values, and their Linear interactions and quadratic effects were estimated. In addition, the coefficients of determination R^2^ and adj-R^2^ were also applied to evaluate the efficiency of the polynomial regression model. These values were identified to be 0.969 and 0.942, respectively, revealing that 96.9% of the variation in MO degradation could be explained by the investigated variables and that only 3.1% of the variation could not be elucidated by these variables. As documented in [74], regression models with high R^2^ values (0.9–1) are deemed to have the strongest and most positive correlations. Higher R^2^ values also indicate that a model is good and, more specifically, that it can explain the variation in experimental data as compared to the predicted data. Furthermore, the adjusted-R^2^ (adj-R^2^) value adjusts the R^2^ value based on the sample size and the number of variables in the model. The adj-R^2^ value should be in considerable agreement with the R^2^ value (≤2%) [75]. In this study, the excellent agreement between R^2^ and adj-R^2^ values verified the aptness of the model. On the other hand, the lack of fit test, which describes the variation in the data around the fitted model, revealed a value of 0.093, as inferred using the ANOVA method (Table 4). Broadly, an insignificant lack-of-fit indicates a good model [56]. Moreover, the model exhibited evident correlation between the observed (experimental) and the predicted values, as inferred from the studentized residual (Table 2). A small residual value (≤ ±2) is accepted. It estimates the error between points and detects outliers. Clearly, the residual values of our model did not deviate from accepted limits and fall on a straight line with only one outlier; this implies that the errors fell within a normal distribution, as demonstrated in the normal probability plots of the residuals (Figure 2). Thus, the model used in this study is optimal for predicting MO detoxification within the range of experimental parameters.

It is noteworthy that the *p*-value is an important tool for assessing the consequences and significance of the model and also each independent variable in the design. Commonly, a low probability ‘*p*’ value (prob > F <0.05) denotes high significance of the corresponding coefficient [56]. In this case, it was found that the ANOVA results, which are tabulated in Table 4, illustrate that the model is highly appropriate and adequate, as evidenced by the very low probability value (0.000). Subsequently, all these data indicated that the model is significant, adequate and well-fitted to the experimental data. Additionally, the linear coefficients of incubation time, inoculum size, and MO and NaCl concentrations were significant for MO degradation with probability values of 0.0, 0.001, 0.0 and 0.016, respectively. However, the quadratic coefficient of the MO concentration, in contrast to the quadratic coefficients of the other variables, was only significant at a level of *p* = 0.0. Generally, the linear effects of the tested parameters had more pronounced effects in terms of enhancing MO degradation than the quadratic and interaction effects, as summarized in Table 4. Additionally, the sign of each coefficient determined either a negative or a positive effect of the corresponding variable on the MO detoxification; specifically, the negative coefficient values of MO concentration (linear effect) implied their negative impact on MO removal at higher levels. However, the positive coefficients of incubation time, inoculum size and NaCl (linear effect) reflected their positive contribution to MO degradation as their values were increased up to certain limits. Similarly, the influence of the interaction between incubation time and NaCl concentration exerted a positive effect on MO degradation (Table 3). Finally, MO degradation as a response could be expressed in the terms of a second-order polynomial equation Equation (7) as following:MO degradation (**%**) = 8.64 + 3.36 incubation time + 2.19 inoculum size − 9.44 MO concentration + 1.39 NaCl concentration + 0.05 (incubation time)^2^ –0.6 (inoculum size)^2^ + 4.0 (MO concentration)^2^ – 0.53 (NaCl concentration)^2^ – 0.82 incubation time × inoculum size – 1.08 incubation time × MO concentration + 1.34 incubation time × NaCl concentration – 2.52 inoculum size × MO concentration–0.396 Inoculum size × NaCl concentration – 0.399 MO concentration MO concentration(7)

### 3.4. Graphical Interpretation of the Response Surface Model

In order to illustrate the relationships between the response and the interactions between the tested variables, three-dimensional (3D) surface plots and two-dimensional (2D) contour plots were generated (Figure 3). The MO degradation was plotted on the Z-axis versus and the two examined parameters were plotted on X and Y axes, while other parameters were kept fixed at their midpoints. The simultaneous effect of incubation time and MO concentration on MO detoxification, at constant level of NaCl concentration and inoculum size, was studied, and the results are presented in Figure 3A,B. It can be seen that more than 40% removal was obtained at the lowest concentration of MO (200 mg/L) under 5 days incubation (the highest incubation time). However, as elucidated in Figure 3C,D, the obvious increase in MO detoxification was conjugated with a mutual increase in both the incubation time and the NaCl concentration, reflecting a synergistic interaction effect. Moreover, an antagonistic relationship between inoculum size and MO concentration was observed, as shown in Figure 3E,F. The highest concentrations of MO resulted in lower MO detoxification within a low range of MO (200–250 mg/L), and high bacterial inoculum sizes (10–20%) assisted in significantly elevating the removal percentage of MO, which reached 50% and above. The same antagonistic trend was also clearly observed in the efficacy of MO degradation as a function of the MO and NaCl concentrations. The maximum degradation of MO (more than 40%) could be achieved at a wide range of concentrations of NaCl (5–40 g) and at an even higher range of MO concentrations (300–400 mg/L) (Figure 3G,H).

To predict the maximum MO degradation under the examined conditions, the reduced regression model was solved using the optimizer tool in MINITAB 14.0, which calculates individual desirability using a desirability function. Such functions define the sufficient combination of variables to gain the maximum response. The result ranges from zero (less than the accepted limit) to one (the accepted limit) [56]. Therefore, the predicted optimal levels of the process variables were as follows: incubation time—5 days; inoculum size—20%; MO concentration—200 mg/L; and NaCl concentration—40 g/L.

### 3.5. Analysis of Degradation By-Products

#### 3.5.1. UV−Visible Spectrophotometric Analysis

As demonstrated in Figure 4A, the changes in the UV–vis spectra (from 200 to 800 nm) of MO were observed upon consortium treatment. The spectrum profile of MO (control) showed the main absorption peak to be at 465 nm, due to the azo bond, which was almost completely vanished in the supernatant of the treated solution. The dissipation of the peak at 465 nm indicates that these consortia were successful in removing MO. Thus, the decolorization of MO by these consortia may be attributed largely to biodegradation. Our results are in agreement with those reported by Akansha et al. [76] and Haque et al. [49].

#### 3.5.2. FTIR Analysis

In order to affirm the complete decolorization and demethylation of anionic MO dye, a comparative investigation of the FTIR spectrum of the supernatant in the presence of dye was conducted before and after the degradation process (Figure 4B,C). Generally, the spectrum of decolorized MO after the completion of the decolorization process displayed significant phenomena such as the disappearance of some characteristic peaks as compared to the control spectrum, the appearance of new bands, and also shifts in existing bands. Figure 4B illustrates the different absorbance peaks assigned to the MO dye molecules, including a peak at 2892 cm^−1^ for asymmetric CH_3_ stretching vibrations, peaks at 1512 and 1410 cm^−1^ for C=C-H in the bend of the C-H plane. Moreover, absorbance peaks were recorded at 1109, 1027, 942 and 821 cm^−1^ for ring vibrations, and a peak was found at 744 cm^−1^ for the disubstituted benzene ring. Remarkably, their presence denoted the aromatic nature of the dye. The peak at 1652 cm^−1^ for the –N=N- stretch and the peaks at 1358 and 1181 cm^−1^ for -C-N revealed the azo nature of the dye. The peaks at 688, 617 and 563 cm^−1^ for the -C-S- stretching vibrations and the S=O stretching vibrations were attributed to the sulfonic nature of the dye [77,78]. The FTIR spectra of MO after the treatment process was found to display a peak at 3212 cm^−1^ for the N-H bend and a peak at 2943 cm^−1^ for the asymmetric -CH_3_ stretching vibrations (Figure 4C). Notably, the absence of peaks at 1594, 1512, 1410, 942, 821, 744, 688 and 617 cm^−1^ reflected the breakdown of aromatic C=C. Additionally, the absence of a peak at 1652 implied the cleavage of the azo bond. Our results are consistent with those reported in [49,76]. However, the appearance of a new band at 1247 cm^−1^ implied the generation of aromatic amines with -C-N vibrations, as highlighted by Masarbo et al. [79] and Baena-Baldiris et al. [80].

### 3.6. Enzymes Activity Assays

A huge number of microbial enzymes have been involved in the decolorization and biodegradation of synthetic dyes. These bioremediation-related microbial enzymes include laccases, oxido-reductases and hydrolases. In the current study, eight biodegradative enzymes produced by the bacterial consortium were assayed in relation to the decolorization of MO as a model of anionic azo dyes. The optimum decolorization time was observed after 96 h of incubation with MO. Bacterial cell lysates exhibited more potent intracellular activities than extracellular supernatant for all tested enzymes. The decolorization of azo dyes such as MO mainly occurred as a result of the biotransformation process, which was mediated by reductive and oxidative enzymes [81]. The results for all tested enzymes (nitrate reductase, azoreductase, DCIP reductase, NADH reductase, lignin peroxidase (LiP), manganese peroxidase (MnP), laccase and tyrosinase) are summarized in Figure 5. The induction of nitrate reductase, azoreductase, DCIP reductase and NADH reductase was estimated to be 21.61 ± 1.96, 23.05 ± 1.74, 306.97 ± 4.78 and 135.985 ± 2.79 IU/min/mg protein, respectively, after incubation periods of 4 days (Figure 5).

The expression of LiP and MnP was found to be 5.53 ± 0.64 and 9.76 ± 1.69 IU/min/mg protein, respectively. Moreover, the expression of laccase was found to be increased dramatically with the increasing of the incubation time, and was estimated to be 1.45 ± 0.43, 5.68 ± 0.43, 10.72 ± 0.57, 19.60 ± 0.91 and 27.63 ± 1.24 IU/min/mg protein after incubation times of 0.5, 1.0, 2.0, 3.0 and 4.0 days, respectively. The expression of tyrosinase was also estimated to be maximally induced after 4.0 days of incubation with enzymatic activity of 52.72 ± 1.06 IU/min/mg protein. Further significance is suggested by the fact that the enzymatic activity of all tested enzymes was found to be increased by the used bacterial consortium in a time-dependent manner throughout the MO decolorization process.

### 3.7. Acute Toxicity Assessment

#### 3.7.1. Phytotoxicity Bioassays with *C. vulgaris*

In medium derived from CCD containing 200 mg/L of MO (control experiment), chlorophyll-a was reduced before treatment with bacterial consortium. The maximum decrease in chlorophyll-a was 62.16% compared to the control (*C. vulgaris* 100%); however, the decrease in chlorophyll-a after treatment with consortium was 14.88% as compared to the control (*C. vulgaris* 100%) (Figure 6A). The decrease in chlorophyll-b was 51.28% before treatment with consortium and 21.49% after treatment with consortium relative to the control (Figure 6B). On the other hand, the carotenoid content was sharply increased before treatment with consortium (218.73%) and sharply decreased after treatment with consortium (22%) (Figure 6C).

#### 3.7.2. Cytotoxicity against Normal Human and Animal Cell Lines

To assess the cytotoxicity of MO and its degradation metabolites on human and animal cells, they were incubated with human normal lung (Wi-38) and monkey kidney (Vero) cells. Then, the viability of these treated cells was determined by means of an MTT assay and by estimation, relative to the untreated healthy cells, based on the formation of insoluble formazan products by active cells. MO dye (200 mg/L) lowered the viability of Wi-38 and Vero cells to 34.10% and 42.57%, respectively. However, upon treatment by immobilized consortium, the remediated solution maintained the viability of both normal cells at 90.63% and 99.23%, respectively (Figure 6D). Accordingly, the results revealed that the treatment with immobilized consortium exhibited most significant potency in terms of reducing the toxicity of MO in both normal cell lines in comparison to dye alone (*p* < 0.0), which reflected the success of MO degradation using the examined consortium.

### 3.8. Bioaugmentation of Free and Immobilized Cells in Real Waste Effluents

The performance of immobilized consortium in comparison to free consortium was examined in real effluent samples, collected from discharge lines of industrial and agricultural zones in Alexandria, Egypt, either solely or in synchronization with indigenous microbiota (i.e., augmentation). The chemical, physical and biological criteria of the examined wastewater were tested and their results are tabulated in Table 5. Some general observations were recorded and listed. Commencing with the non-immobilized alginate beads, which were run in parallel as a control to examine the absorption, an uptake percentage ranging from 10.82 to 12.88% was recorded. Furthermore, the biodegradation process was time dependent; specifically, higher biodegradation was implemented gradually as a function of the time, as shown in Figure 7. In this process, the immobilized consortium degraded 75.5 and 61.5 mg/L within 72 h in the augmented system of agricultural and industrial effluent, respectively; however, at 144 h., 155 and 128.5 mg/L were remediated by immobilized consortium in the augmented system of agricultural and industrial effluent, respectively. Meanwhile, more efficient biodegradation of MO was observed in both augmented systems, either free or immobilized, as compared to the non-bioaugmented systems. Increments in MO removal of about 13.53 and 8.64% in the augmented systems of the agricultural and industrial effluents, respectively, were attained with the use of immobilized consortium. However, for the freely-added consortium, 8.59 and 5.43% increments in MO elimination were recorded in the agricultural and industrial effluents’ augmented systems, respectively. Notably, the immobilized consortium exhibited higher biodegradation potential than free consortium, with 77.59 and 62.38% removal in the bioaugmented system of agricultural effluent; for comparison, 69 and 53.74% removal were attained in the non-bioaugmented system. Regarding the industrial effluent, 64.78 and 51.38% removal was obtained with immobilized and free augmented consortium, respectively; in the non-bioaugmented system, the immobilized and free consortium eliminated 59.35 and 37.85% of MO, respectively. Broadly, the MO bioremediation efficiency in agricultural wastewater displayed better results than those recorded for industrial effluent (Figure 7).

## 4. Discussion

The discharge of dye effluents into water bodies is becoming a real threat. During the dying process, about 10–70% unfixed dye is disposed, causing substantial ecological deteriorations such as damaging the photosynthesis process, changing the water quality and bringing about adverse aesthetic influences. These issues are further compounded by the gradual accumulation of dye in the tissues of aquatic creatures, which subsequently causes serious issues for human health through the food chain [82]. In the present study, the immobilized microbial consortium was employed to bioremediate MO in a real simulation model. The process began with the accurate selection of microbes that possess biodegradative capabilities. The selected strains *Raoultella planticola*, *Ochrobactrum thiophenivorans*, *Bacillus flexus* and *Staphylococcus xylosus* exhibited promising biodegradation traits within time range of 20–28 h., either solely or combined in a consortium, achieving complete degradation in 18 h with the use of NB medium. Interestingly, the presence of compatibility and mutually beneficial corporation in the mixed co-culture system accelerates the bioremediation process, as mentioned by Weihua Tang et al. [83].

Notably, numerous dyestuff degradation studies focused on the testing of degradation and decolorization using highly nutritive microbiological ingredients [49,69,76,84]. In fact, peptone and yeast extract were widely used for this purpose as carbon sources, nitrogen sources or both. Both trigger the generation of NADH, which acts as redox mediator during the metabolization process, and thus, aids in energy production, survival and multiplication of microorganisms. Nonetheless, their presence may cause only decolorization with incomplete decomposition. In fact, the microorganisms first utilize easily metabolized rich nutrients, rather than assimilating recalcitrant compounds as dye, in order to survive and proliferate [49]. Additionally, the effluents are not always full of easily metabolized nutrients such as complex microbiological media. However, effluents may be poorly nutritive and supplemented with various impurities such as surfactants, pesticides and biocides, which are relatively often accompanied by high salinity [79]. This directed us to examine the assimilation behavior of consortium toward MO in MSM that lacked complex nutrients, in a way that mimicked what happens in reality. In this case, the bacterial consortium was imposed to decompose MO into simpler structures for growth and multiplication. Remarkably, the combinatory state of microbial species could permit full degradation. In fact, individual bacterial species could utilize MO incompletely and its intermediate byproduct could be consumed by another bacterial species in such mixed-culture consortiums; ultimately, full decomposition of the dye was achieved without the accumulation of hazard residues [49]. Different studies supported our finding regarding MO assimilation [85]. However, the utilization of freely suspended biomass for the purpose of degradation of toxic materials, especially in situ, would restrict its efficacy. This fact could be attributed to the low mechanical strength of microbial cells at this state, as well as their susceptibility to harsh environmental conditions and toxic byproducts, which would alternate their metabolic activities. To circumvent such limitations, the immobilization approach was employed. Such an approach provides stability and protection for microbial cells, and ensures viability over a prolonged period. Importantly, it also permits in situ applications and recovery/reuse of microbial cells [24,86]. Thus, the entrapment of the consortium in ionotropic hydrogels such as alginates would lead to the promotion of the bioremediation process. The natural origin of alginate, its biocompatibility, low toxicity, easy gelling operation and inexpensiveness encourage its applications in biotechnological, medical and environmental usages. Furthermore, the research presented in [87] reinforces this point of view. In this study, it was shown that the entrapment of cells in a suitable matrix would increase the retention time of the cells in wastewater; consequently, the treatment process would be simplified.

It is plausible to study the effects of different parameters on MO’s degradation performance. Several levels of various parameters were tested simultaneously in a systematic, cost-effective, time-saving approach, which was attained via CCD. In fact, the conventional single-dimensional “one variable at time (OVAT)” strategy is a time consuming, laborious practice due to large number of required experiments and often fails to explain the interaction effects among the multivariable parameters [56,75]. Through the statistical design of the experiment (CCD), different levels of dye concentrations, NaCl concentrations, and inoculated sizes were studied in 31 trials within a time range of 1–5 days. The statistical model revealed that a 20% inoculation of the biomass with the immobilized consortium effectively eliminated 200 mg/L of MO in the presence of 40 g/L of NaCl within 5 days of incubation. Interestingly, our immobilized consortium was characterized by its ability to remove MO under high salinity. On the other hand, Weihua Tang et al. highlighted the importance of the salt tolerance property in the dye decolorization process, especially from wastewater [83]. He and coworkers referred to the removal efficiency of the fungus-algae consortium to decolorize 58.24% of dye from simulated wastewater in the presence of 20 g/L of NaCl. In addition, Akansha et al. recorded the decline in MO decolorization by *Bacillus stratosphericus* SCA1007 in the salinity range of 4–7%, attributing this case to cell plasmolysis [76]. Moreover, Haque et al. pointed out the ability of biofilm consortium to detoxify 68.34% of MO in a highly nutritive peptone medium yeast extract (YEP) containing 10% NaCl [49]. Thus, it could be deduced that our consortium is promising in terms salt-tolerant azo degradation and that it has significant potential to be applied in high salinity environments.

Remarkably, the existence of adequate inoculum sizes can enhance dye-treatment operations. As noted by Sari and Simarani [66], the performance of the *L. fusiformis* strain W1B6 in azo dye decolorization increased with the elevation of the inoculum size from 5% to 10%. Furthermore, Haque et al. stated that the azo dye molecules obstructed the active azoreductase sites of lower biomass ratios of inoculated consortium, reflecting the significance of microbial biomass availability in terms of improving dye bioremediation via the donation of more azoreductase active sites [49]. In the same context, the decolorization performance of azo dye was inversely proportional to the initial dye concentration [83]. At higher dye concentrations, the decolorization performance declined due to the saturation of degradative enzymes’ active sites with dye molecules, the inactivation of the transport system and the overcrowding of dye byproducts, which ultimately led to the inhibition of microbial metabolism. In agreement with our results, Haque et al. documented that the MO removal efficiency and also its growth rate were significantly diminished by increasing the MO concentration from 100 to 400 mg/L [49]. The exact observation was recorded elsewhere [66,88]. On the other hand, the biodegradation is directly proportional to the incubation time. In fact, MO removal was ameliorated within the examined time course, which was in line with the results obtained by Haque et al. [49], who found that MO removal increased from 59.34 to 79.21% with the increasing of the incubation time from 36 to 48 h. Broadly, dyeing wastewater always contains high salt and dye concentrations of 3–10% and 10–200 mg/L, respectively [49,89]. Therefore, for environmental applications, the ability of microorganisms to withstand high concentrations of dye and salinity is an indispensable property in terms of guaranteeing successful bioremediation.

Based on results of the enzyme assay regarding the synchronization of the UV-Vis spectra and the FTIR of the biodegraded solution, the mechanism of the degradation process could be speculated upon. The absence of both orange color from remediated solution and also the absorption maxima at 465 nm indicated complete cleavage of the N=N bond, as highlighted by Costa et al. [12]. Such results were affirmed by FTIR via the absence of the azo bond at 1652 cm^−1^, which signified the initiation of the biodegradation process by the action of azoreductase, which mediated the reductive disputing of the N=N bond. However, the disappearance of some peaks unveiled the breakdown of the benzene ring by the dint of DCIP reductase and NADH reductase, which were considered to be key enzymes for biodegradation in the current study. Nonetheless, other oxidative enzymes, such as LiP, MnP and lacasse, played important roles in the accomplishment of the MO bioremediation process. They mediated the oxidation of aromatic and inorganic compounds and their mineralization into simpler structures such as 1, 4-benzene diamine and N, N-dimethyl, as inferred from the appearance of new peaks in FTIR. Interestingly, several mechanisms conducted MO degradation enzymatically, either through non-specific free radical mechanisms, as was the case for laccases, or through free-radical production that oxidizes azo dyes, as was observed for LiP, MnP and peroxidases. Furthermore, the non-enzymatic route of degradation involves low molecular weight redox mediators and biogenic reductants that mediate chemical reduction [90]. Our results agreed with those reported by Haque et al. [49] who documented the induction of azoreductase, NADH-DCIP reductase and laccase by their biofilm consortia for MO decolorization in a similar time frame. Additionally, Ayed et al. [91] found that the microbial consortium of *Sphingomonas paucimobilis, Bacillus cereus*ATCC14579 and *Bacillus cereus*ATCC11778 decolorized MO (>84%) within 48 h under optimal conditions. They revealed a significant increase in the expression of azoreductase, LiP and laccase. On the other hand, Sheela and Sadasivam noticed an increase in laccase level expression caused by *Bacillus cereus* SKB12 during the biodegradation of textile dyes, while the level of expression of the NADH-DCIP reductase and MnP enzymes was found to be poor [92]. Furthermore, Thanavel et al. revealed that textile dye degradation was successfully implemented by a strain of *Aeromonas hydrophila* SK16, which showed a significant expression of degradative enzymes such as tyrosinase, LiP, riboflavin reductase, azoreductase and laccase with the increasing of their levels during the degradation process [93]. However, the decolorization and cleavage of azo dye did not ensure the safety and nontoxicity of metabolic end products; thus, a toxicity study should accompany these results. Generally, phytotoxicity was examined through the seed germination of plants such as *Vigna radiata* [76], *Triticum aestivum* and *Sorghum vulgare* [49,79], and *Phaseolus mungo* and *Sorghum vulgare* [94]. Herein, the phytotoxicity assay was utilized to assess the impact of the original dyes’ and metabolites’ outcomes on the growth and photosynthesis performance of *C. vulgaris*. As it is a photosynthetic unicellular microalgae, it is characterized by its rapid growth and its sensitivity to toxic compounds; hence, it could provide a quick insinuation of the effect of bioremediated wastewater on photosynthesis and could possibly be applied in the subsequent irrigation of plants [95]. The phytotoxicity results revealed the toxic effect of MO (200 mg/L) and the inhibition of the photosynthesis process as compared to the control, which were deduced from the diminishing of chlorophyll-a and chlorophyll-b and the increase in carotenoid content. However, the impact of bioremediated water on the algae photosynthesis process was much lower than that observed in the original MO (200 mg/L) due to the lowering of the chlorophyll-a and chlorophyll-b contents to 14.88 and 21.49% and the lowering of the carotenoid content to 22% The discrepancy in the behaviors of the chlorophyll and the carotenoids in response to either MO or its metabolic byproducts could be attributed to the antioxidant activity displayed by carotenoids under stress conditions in order to tolerate and adapt to hazardous compounds [96,97].This is consistent with the findings of Liao et al. [98], who documented that the EC50 of azo dye, Reactive Black B (RBB), was 48 mg/L against the growth of the freshwater algal *C. vulgaris*, assigned this result to the malfunctioning of chloroplasts. On the other hand, the cytotoxicity of MO and its bioremediated byproduct against the human cell lines Wi-38 and Vero cells ensured the safety and reduced toxicity of MO-bioremediated water via the maintenance of the viability of both normal cells at 90.63% and 99.23%, respectively. Broadly, our toxicity results were promising, suggesting the safe employment of MO-bioremediated wastewater, through the use of the immobilized consortium under study, for ferti-irrigation purposes to overcome water crisis issues.

To emphasize the effectiveness of the consortium under study and compare its efficiency in different effluents, it was employed in the bioremediation of real agricultural and industrial wastewater supplemented artificially with 200 mg/L of MO, using augmented and non-augmented systems, in both free and immobilized states. As noticed, the water quality parameters were analyzed and indicated some common differences between both effluents, which would aid in explaining and perceiving the overall performance of the bioremediation process. The agricultural effluents yielded higher values in terms of total count, biological oxygen demand (B.O.D), sulfate, carbonate, calcium, total nitrogen and phosphate contents as compared to the industrial effluents. That could be ascribed to the presence of fertilizers, nitrogenous/phosphorous nutrients, soil stabilizers, herbicides and pesticide-containing residues. On the other hand, the industrial effluents yielded higher amounts of oil, phenolic compounds, chemical oxygen demand (C.O.D) and several heavy metals. By applying our microbial consortium either as free or immobilized in these real effluents, a difference in bioremediation performance was observed according to the method of application, namely the augmented and the non-augmented system, and also the nature of effluents. As stated by Qu et al. [99] and Zommere et al. [86], the success or failure of a bioaugmentation process depends substantially on the activity and persistence of the introduced consortium in addition to its competitive interaction with indigenous microbiota in the contaminated area. Despite simplicity and success at the laboratory scale, several studies reported failures in the bioaugmentation operation owing to the previously mentioned reasons [86,100]. In the current research, our mixed-culture consortium achieved the biodegradation and noticeable removal of MO in augmented systems of both examined effluents, reflecting its corporative interaction with native indigenous microorganisms without any competitive inhibition despite the variations in the natures and contents of the two different wastewaters. However, the observed degradative capability of the consortium in the non-augmented systems of both wastewaters revealed its ability to withstand harsh circumstances (inhibitory substances such as heavy metals, phenols, oil, nitrogen/phosphorous content, osmotic stress, etc.). Furthermore, the higher removal capacity of the immobilized state compared to the freely suspended state could be attributed to the presence of microbial cells in the highly diffusive protective matrix, which provided better contact with MO due to its absorption capability, without direct contact with other toxic pollutants and in a protective manner. This result concurred with that given in [24]. Meanwhile, the bioremoval potency of the immobilized consortium in agricultural wastewater was higher than that observed in industrial effluent. This could be assigned to the presence of more stressor substances such as heavy metals, oils and phenols. Such contents could have a negative influence on the degradative performance of the consortium. In addition, the presence of a higher microbial count in agricultural effluent, as a result of the higher contents of nutritive substances, could assist in achieving improved degradation of MO. Generally, the mutual and synergistic interaction among the introduced consortium and indigenous microorganisms enhanced the decolorization and degradation of MO in this bioaugmentation operation. It is plausible to mention that the prolongation of the incubation time would result in a progressive enhancement in the bioremediation process, which is consistent with the findings of Jublee Jasmine and Suparna Mukherji [101] and Poi et al. [100]. The promising results of the bioaugmentation operation with our characteristic immobilized consortium encourage us to study the dynamics of bioaugmentation processes that are specifically designed for a wider variety of bioaugmented systems, in order to meet industrial requirements and comply with environmental legislation.

## 5. Conclusions

The current study aimed to construct an efficient bacterial consortium of *Raoultella planticola*, *Ochrobactrum thiophenivorans*, *Bacillus flexus* and *Staphylococcus xylosus* to degrade MO. The consortium exhibited considerable ability to assimilate MO as a carbon source in mineral salt media with stronger efficiency than bacteria alone. The degradation behavior of the immobilized consortium using CCD indicated its efficacy in degrading 200 mg/L of MO under high salinity (40 g/L of NaCl) within 5 days of incubation. The degradation potency was emphasized by the expression of degradative enzymes such as azoreductase, DCIP reductase, NADH reductase, laccase, LiP, MnP and tyrosinase. The UV−Visible spectrophotometric analysis and FTIR examination of degradation outcomes revealed the breakdown of the azo bond and the aromatic groups. The toxicity study of such metabolic byproducts was assessed by determining their influence on the photosynthetic performance of *C. vulgaris* and the viability percentage of human cell lines (Wi-38 and Vero cells). Moreover, the immobilized consortium, in comparison to freely suspended culture, was employed in the bioremediation of MO, from artificially contaminated agricultural and industrial effluents, in augmented and non-augmented systems. The results highlighted the mutually corporative interaction between immobilized consortium and indigenous microorganisms in the biodegradation of MO in both examined effluents, reflecting the adaptation capacity of our consortium to effluents of various kinds and from various sources.

## Figures and Tables

**Figure 1 biology-11-00076-f001:**
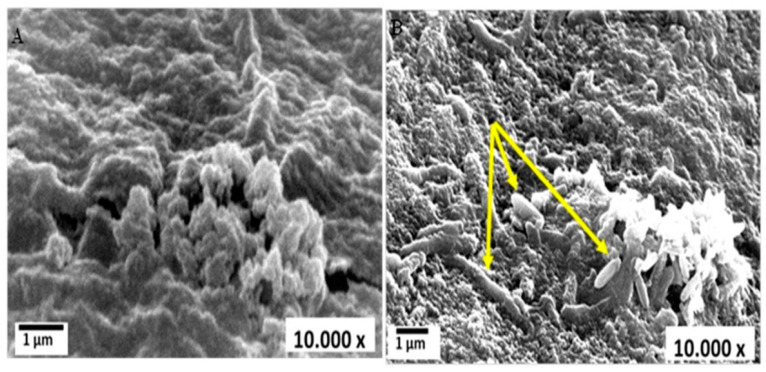
Scanning electron microscope (SEM) micrograph of blank alginate microsphere before and after bacterial cell immobilization: (**A**) blank alginate microsphere; (**B**) immobilized bacterial consortium in alginate microsphere.

**Figure 2 biology-11-00076-f002:**
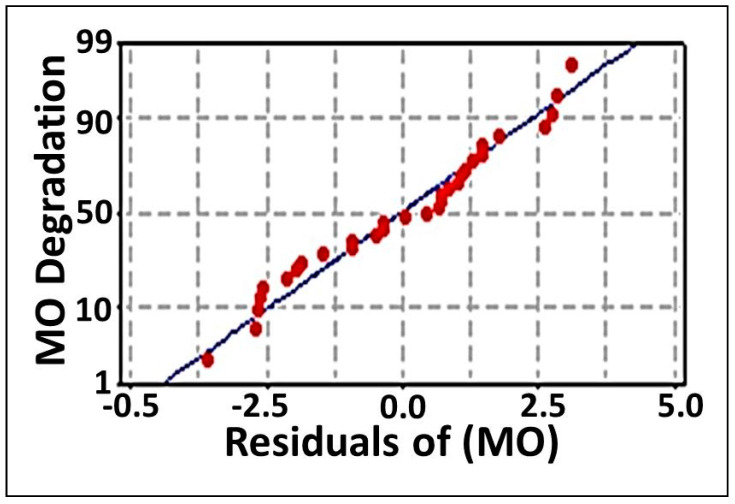
The normal probability plot of the residuals for MO degradation by immobilized consortium determined by the second-order polynomial equation.

**Figure 3 biology-11-00076-f003:**
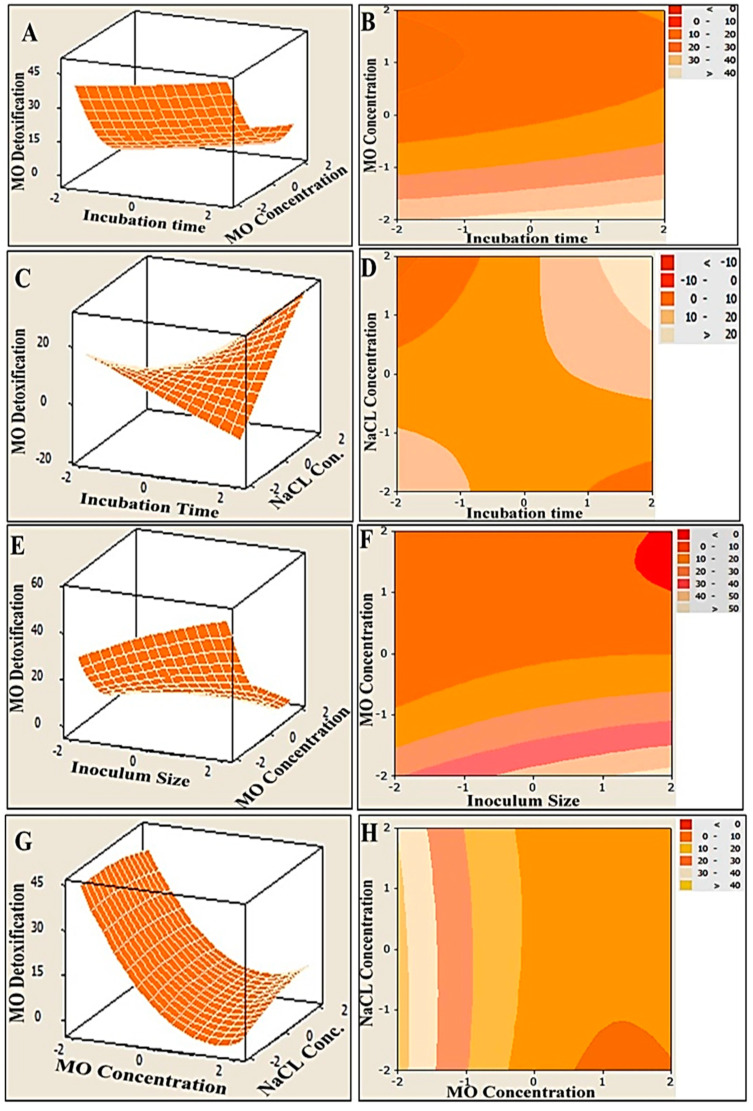
Three-dimensional surface and two-dimensional contour plots of MO degradation by immobilized consortium, showing the interactive impact of two variables at constant zero levels of the other variables. Both plots were created using MINITAB-14 software. (**A**,**B**) are the effect of incubation time and MO concentration on MO detoxification, (**C**,**D**) show obvious increase in MO detoxification with a mutual increase in both incubation time and NaCl concentration, (**E**,**F**) show antagonistic relationship between inoculum size and MO concentration, (**G**,**H**) show the maximum degradation of MO at a wide range of concentrations of NaCl and high range of MO concentrations.

**Figure 4 biology-11-00076-f004:**
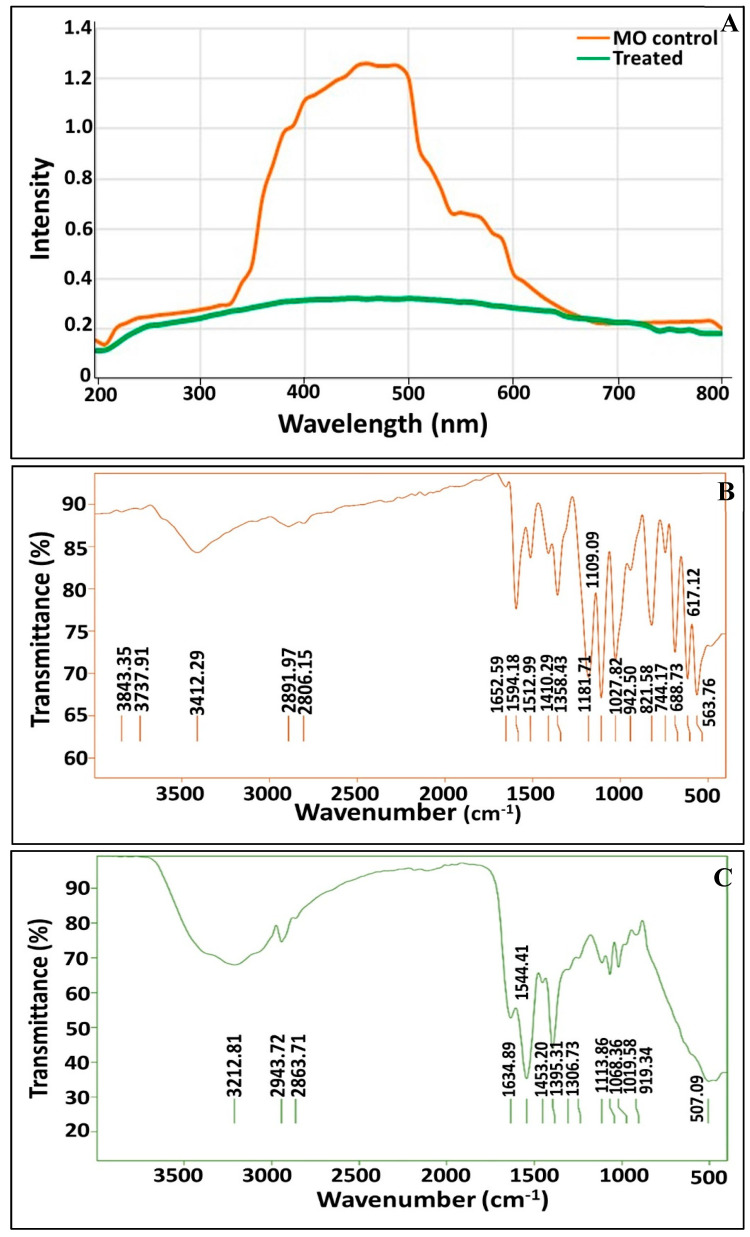
UV-Vis spectroscopy profile of MO before and after treatment by immobilized consortium (**A**); FTIR spectra of MO before degradation (**B**) and after degradation by immobilized consortium (**C**).

**Figure 5 biology-11-00076-f005:**
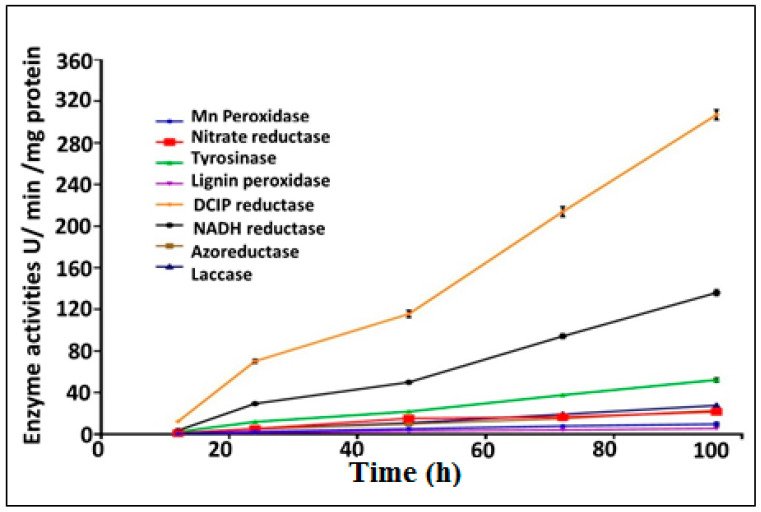
Quantitative bio-induction of eight different degrading enzymes during the decolorization of MO dye by immobilized consortium.

**Figure 6 biology-11-00076-f006:**
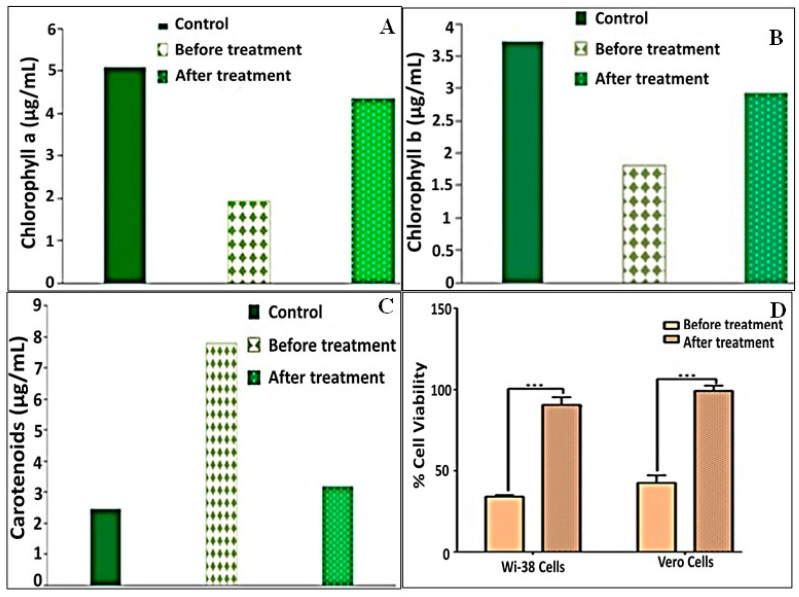
Phytotoxicity of methyl orange (200 mg/L) and remediated byproducts against *C. vulgaris* content of (**A**) chlorophyll-a, (**B**) chlorophyll-b and (**C**) carotenoids. (**D**) Cytotoxicity against normal (human Wi-38 and mammalian Vero) cell lines in terms of cell viability.

**Figure 7 biology-11-00076-f007:**
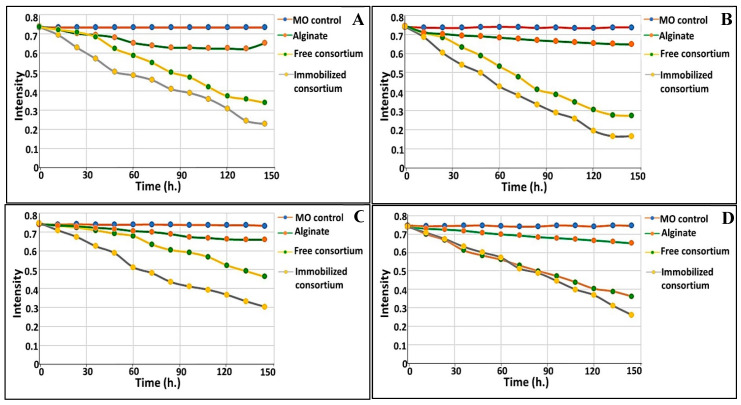
MO bioremediation from agricultural and industrial effluents using microbial consortium either freely suspended or immobilized in both augmented and non-augmented states: (**A**) non-augmented state of agricultural effluent; (**B**) augmented state of agricultural effluent; (**C**) non-augmented state of industrial effluent; (**D**) augmented state of industrial effluent.

**Table 1 biology-11-00076-t001:** The concentrations of examined variables (incubation time, inoculum size, MO conc. and NaCl) at each level in the CCD experiment.

Variable	Coded Levels/Experimental Values				
	−2	−1	0	1	2
Incubation time (days)	1	2	3	4	5
Inoculum Size (%)	1	3	5	10	20
MO Conc. (mg/L)	200	250	300	350	400
NaCl Conc. (g/L)	5	10	15	20	40

**Table 2 biology-11-00076-t002:** Central composite design (CCD) matrix of MO degradation using immobilized consortium influenced by incubation time, inoculum size, MO and NaCl concentrations along with the predicted responses and residuals.

Run Order	Incubation Time (Day)	Inoculum Size	MO Conc.	NaCl Conc.	Experimental MO Degradation (%)	Predicted MO Degradation (%)	St. Residual
1	0	0	2	0	2.25	5.825	−2.17
2	0	0	−2	0	46.38	43.61	1.68
3	−1	1	−1	−1	25.14	22.029	1.89
4	−1	−1	−1	1	11.37	11.834	−0.28
5	0	0	0	0	8.31	8.641	−0.14
6	−2	0	0	0	0.27	2.122	−1.13
7	−1	−1	1	1	0.52	−0.638	0.7
8	1	−1	−1	1	22.4	25.081	−1.63
9	0	0	0	0	11.5	8.641	1.21
10	0	0	0	−2	1.1	3.7	−1.58
11	0	−2	0	0	2.81	1.7	0.67
12	1	1	1	1	5.24	5.171	0.04
13	0	2	0	0	8.58	10.495	−1.16
14	1	−1	1	−1	3.49	2.771	0.44
15	0	0	0	0	9.11	8.641	0.2
16	−1	1	1	−1	2.36	1.067	0.79
17	1	1	−1	1	32.77	32.082	0.42
18	0	0	0	2	11.08	9.285	1.09
19	1	−1	1	1	7.34	8.257	−0.56
20	2	0	0	0	16.62	15.574	0.64
21	0	0	0	0	8.31	8.641	−0.14
22	1	1	1	−1	3.93	1.272	1.62
23	−1	1	−1	1	20.02	22.128	−1.28
24	−1	−1	1	−1	0.78	−0.726	0.92
25	1	1	−1	−1	24.04	26.586	−1.55
26	0	0	0	0	7.75	8.641	−0.38
27	0	0	0	0	6	8.641	−1.12
28	−1	−1	−1	−1	8.69	10.148	−0.89
29	−1	1	1	1	1.04	−0.431	0.89
30	1	−1	−1	−1	18.72	17.997	0.44
31	0	0	0	0	9.51	8.641	0.37

**Table 3 biology-11-00076-t003:** Estimated effect, regression coefficients and corresponding *T* and *p* values for determining the effects of different variables on MO degradation by means of immobilized consortium using CCD.

Term	Coef.	SE Coef.	T	*p* Value
Constant	8.64143	0.9636	8.968	0
Incubation days	3.36292	0.5204	6.462	0
Inoculum size	2.19875	0.5204	4.225	0.001
MO conc.	−9.44625	0.5204	−18.152	0
NaCl conc.	1.39625	0.5204	2.683	0.016
(Incubation days)^2^	0.05162	0.4768	0.108	0.915
(Inoculum size)^2^	−0.63588	0.4768	−1.334	0.201
(MO conc.)^2^	4.01912	0.4768	8.43	0
(NaCl conc.)^2^	−0.53713	0.4768	−1.127	0.277
Incubation days × Inoculum size	−0.82313	0.6374	−1.291	0.215
Incubation days × MO conc.	−1.08812	0.6374	−1.707	0.107
Incubation days × NaCl conc.	1.34938	0.6374	2.117	0.05
Inoculum size × MO conc.	−2.52188	0.6374	−3.957	0.001
Inoculum size × NaCl conc.	−0.39687	0.6374	−0.623	0.542
MO conc. × NaCl conc.	−0.39937	0.6374	−0.627	0.54

**Table 4 biology-11-00076-t004:** Analysis of variance (ANOVA) for MO degradation by means immobilized consortium obtained using CCD.

Source	Df	Seq. SS	Adj. SS	Adj. MS	*F*	*p* Value
Regression	14	3259.04	3259.04	232.788	35.82	0
Linear	4	2575.8	2575.8	643.949	99.07	0
Square	4	517.49	517.49	129.373	19.9	0
Interaction	6	165.75	165.75	27.625	4.25	0.01
Residual error	16	103.99	103.99	6.5		
Lack of fit	10	86.86	86.86	8.686	3.04	0.093
Pure error	6	17.14	17.14	2.856		
Total	30	3363.03				

**Table 5 biology-11-00076-t005:** Quality analysis of agricultural and industrial effluents.

Parameter	Industrial Wastewater Concentration	Agricultural Wastewater Concentration
Total nitrogen (mg/L)	113.44	411.2
Total phosphates (mg/L)	0.2	194.6
T.D. (mg/L)	1060	927
Nitrate (mg/L)	18.77	274.9
Nitrite (mg/L)	2.5	44.9
Ammonia (mg/L)	0.135	160.6
Carbonate (mg/L)	81.6	84
Sulfate (mg/L)	133.7	424.58
Sulfide (mg/L)	198.6	58.9
Phenol (mg/L)	73.5	0.04
Oil (mg/L)	185.2	13.3
B.O.D (mg/L)	120.8	183
C.O.D (mg/L)	310	204
Turbidity (NTU)	464.3	12.5
E.C (μs)	1003	901
Total count (CFU/L)	8.34 × 10^3^	6.74 × 10^5^
Calcium (mg/L)	152.52	440.68
Zn (mg/L)	13.4	1.2
Fe (mg/L)	28.6	9.8
Cr (mg/L)	6.6	<0.01
Cd (mg/L)	3.5	<0.01
Cu (mg/L)	11.2	2.3
Co (mg/L)	7.4	0.1
Ag (mg/L)	5.3	<0.01
Pb (mg/L)	9.6	<0.01

## Data Availability

Not applicable.

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
