# Peer review of "Methyl Orange Biodegradation by Immobilized Consortium Microspheres: Experimental Design Approach, Toxicity Study and Bioaugmentation Potential"

_biology, 2022, doi:10.3390/biology11010076_

Round 1

Reviewer 1 Report

  1. Authors should unify the units according to SI standard
  2. Degrees should be written without space between degree symbol and value
  3. Authors should unify the degree symbol as sometimes its o rather than an appropriate symbol
  4. SEM images are completely unreadable 
  5. All figures must be added in higher resolution
  6. Functional Groups like CH3 should have a lower or higher index
  7. Fig 4 A, x-axis should start from 190 rather than 90 for better viewing
  8. Fig 7 - x-axis should be stretched because 120 and 150 minutes are off
  9. on the 21st page has 3 different fonts and size
  10. How do authors check the cell density?
  11. Add names and models of used equipment
  12. Figure 7 and throughout the text change the abbreviation hrs to h
  13. Please add the citation for a better understanding of sorption and removal of dyes: https://doi.org/10.3390/ma14237482

Author Response

# Reviewer 1 comments are in Bold and underlined:

We are grateful and appreciate your revision. We revised all manuscript and the correction was performed also, all of the following points have been clarified.

C1: Authors should unify the units according to SI standard.

A1: Thanks for your deep revision. The required modification has been done.

C2:  Degrees should be written without space between degree symbol and value

A2: It was unintended mistake. We correct it to according to your suggestion

C3:      Authors should unify the degree symbol as sometimes its o rather than an appropriate symbol

A3 : The correction was performed according to your recommendation.

C4: 4. SEM images are completely unreadable

A4: The SEM images resolution were improved. For declaration, the alginate polymer exhibited unspecific shape, however, upon immobilization the bacterial cells appeared clearly as pointed out by arrows.

C5: All figures must be added in higher resolution.

  A5: Thanks for your thoroughly revision. The quality of figures has been improved and added in higher resolution (300 dpi).

C6: 6.  Functional Groups like CH3 should have a lower or higher index.

A6: The required modification has been done.

C7: 7.  Fig 4 A, x-axis should start from 190 rather than 90 for better viewing.

           A7: The suggested correction was done.

C8: Fig 7 - x-axis should be stretched because 120 and 150 minutes are off

  A8: Thanks for your observation. The correction was amended.

C9: on the 21st page has 3 different fonts and size.

           A9: The correction was performed according to your recommendation.

C10: How do authors check the cell density?

           A10: Thanks so much. We use a spectrophotometer to check optical density (OD) measurements.

C11: Add names and models of used equipment.

           A11: We followed your recommendation and added all required data.

C12: Figure 7 and throughout the text change the abbreviation hrs to h.

           A12: We correct it.

C13: Please add the citation for a better understanding of sorption and removal of dyes: https://doi.org/10.3390/ma14237482.

A13: We followed your recommendation and added the suggested reference.

Reviewer 2 Report

Dear Authors

The degradation of methyl orange (MO) which is a more refractory molecule among xenobiotics makes this manuscript extremely interesting in the field of bioremediation, however the authors must clarify: 1- it is possible to create bioreactors or water treatment equipment for a large-scale bioremediation of the OM or these represent only basic non-applicative studies;
2- The molecules that are obtained from the degradation of MO are not reported. However, it cannot be established if there are compounds obtained from the degradation that may represent molecules that are still dangerous for the environment and for human health.
3-The authors must clarify why they preferred the FT-IR technique to that of gas-chromatography (GC-MS) which is usually preferred because
the GC-MS tecnique allows us more accurate quantizations in these kind of determinations;

Best Regards

Author Response

# Reviewer 2 comments are in Bold and underlined:

The degradation of methyl orange (MO) which is a more refractory molecule among xenobiotics makes this manuscript extremely interesting in the field of bioremediation.

We are grateful and appreciate your positive feedback

The Referee has brought up some constructive suggestions and we appreciate the opportunity to clarify our research objectives and results. As indicated below, we have checked all the general and specific comments pointed out.

C1: it is possible to create bioreactors or water treatment equipment for a large-scale bioremediation of the OM or these represent only basic non-applicative studies.

A1: Thanks for your comment. We designed the plane of work to be applied in reality otherwise it was useless. That was performed through examining the consortium in immobilized state and also in our application in real waste water samples for examining the efficiency of our consortium in association with different indigenous microbiota in different effluents. Our results are promising indicated by the compatibility of our consortium with different indigenous microbes and its efficiency of degradation was exhibited despite various contamination nature of examined effluents. Regarding to the bioreactor stage, it is actually running study nowadays in our lab (ongoing study) based on the results of the current manuscript. This reflects that we can use this consortium of bacteria in a large scale of the environment to break down many harmful wastes, including dyes.

C2:  The molecules that are obtained from the degradation of MO are not reported. However, it cannot be established if there are compounds obtained from the degradation that may represent molecules that are still dangerous for the environment and for human health.

A2: Appreciating and agreeing with your vision. However, we performed the toxicity study to confirm the nontoxicity nature of degradation byproducts. As observed in results, higher toxicity was reported for MO before degradation. While upon degradation its metabolic byproducts became less dangerous and nontoxic as reflected by algae growth and its pigment contents (Phytotoxicity study) and also viability of normal human cell lines (Cytotoxicity).

C3:The authors must clarify why they preferred the FT-IR technique to that of gas-chromatography (GC-MS) which is usually preferred because the GC-MS tecnique allows us more accurate quantizations in these kind of determinations.

A3 : Thanks so much. As stated at the discussion of the manuscript  concerning the FTIR results after MO degradation that many peaks were disappeared after the degradation not only the peak at 1500 cm–1  but also, it was noticed the absence of peaks at 1594, 1410, 942, 821, 744, 688, 617 cm–1   that confirm the breakdown of aromatic C=C. Also, the absence of a peak at 1652 implied the cleavage of azo bond. Unfortunately, the availability to support the FTIR results with GC-MS technique is very difficult as the equipment is not present at our universities or even our governates. Additionally, many published studies utilized FTIR to confirm the degradation of azo dyes such as (Haque et al., 2021).

Haque, M. M., Haque, M. A., Mosharaf, M. K., & Marcus, P. K. (2021). Decolorization, degradation and detoxification of carcinogenic sulfonated azo dye methyl orange by newly developed biofilm consortia. Saudi Journal of Biological Sciences, 28(1), 793–804. https://doi.org/10.1016/j.sjbs.2020.11.012